## Use of routine death and illness surveillance data to provide insight for UK pandemic planning: lessons from COVID-19

Helen E Clough  ,[1] K Marie McIntyre,[1] Grace E Patterson,[1] John P Harris,[2] Jonathan Rushton[1]

¹Department of Livestock and One Health, Institute of Infection, Veterinary and Ecological Sciences, University of Liverpool, Neston, UK
²Department of Public Health and Policy, Institute of Population Health Sciences, University of Liverpool, Liverpool, UK

**Correspondence to**
Dr Helen E Clough;
h.e.clough@liverpool.ac.uk

## ABSTRACT

**Objectives** Reporting of COVID-19 cases, deaths and testing has often lacked context for appropriate assessment of disease burden within risk groups. The research considers how routine surveillance data might provide initial insights and identify risk factors, setting COVID-19 deaths early in the pandemic into context. This will facilitate the understanding of wider consequences of a pandemic from the earliest stage, reducing fear, aiding in accurately assessing disease burden and ensuring appropriate disease mitigation.

**Setting** UK, 2020.

**Participants** The study is a secondary analysis of routine, public domain, surveillance data and information from Office for National Statistics (ONS), National Health Service (NHS) 111 and Public Health England (PHE) on deaths and disease.

**Primary and secondary outcome measures** Our principal focus is ONS data on deaths mentioning COVID-19 on the death certificate. We also consider information provided in NHS 111 and PHE data summaries.

**Results** Deaths with COVID-19 significantly contributed to, yet do not entirely explain, abnormally elevated all-cause mortality in the UK from weeks 12–18 of 2020. Early in the UK epidemic, COVID-19 was the greatest threat to those with underlying illness, rarely endangering people aged under 40 years. COVID-19-related death rates differed by region, possibly reflecting underlying population structure. Risk of COVID-19-related death was greater for healthcare and social care staff and black, Asian and minority ethnic individuals, having allowed for documented risk factors.

**Conclusion** Early contextualisation of public health data is critical to recognising who gets sick, when and why. Understanding at-risk groups facilitates a targeted response considering indirect consequences of society's reaction to a pandemic alongside disease-related impacts. COVID-19-related deaths mainly mirror historical patterns, and excess non-COVID-19-related deaths partly reflect reduced access to and uptake of healthcare during lockdown. Future outbreak response will improve through better understanding of connectivity between disease monitoring systems to aid interpretation of disease risk patterns, facilitating nuanced mitigation measures.

### Strengths and limitations of this study

► The study shows how routine, public domain data can be used to provide pertinent insight into a pandemic in its earliest stages.
► The use of imaginative approaches to graphical display and numerical commentary ensures that the work can be understood by readers without a statistical specialism.
► This study uses a freely available statistics package to explore public domain data sets, ensuring that results are both transparent and repeatable.
► Insight is limited by problems in identifying raw data from some sources: improving ease of access will strengthen this process and improve the relevance of future inferences.
► Inference is currently restricted to the UK, but the same process could be applied in other countries.

## INTRODUCTION

Intense media reporting during the COVID-19 pandemic has focused on presenting daily data on cases, deaths and testing associated with the virus. The pandemic has undoubtedly changed our world—governments have employed unprecedented (in our times) lockdown methods to reduce transmission. These measures have greatly impacted society. The regular reporting of daily COVID-19 infections and deaths has alarmed the public, particularly when understanding of risk factors dictating severity of COVID-19 symptoms is only slowly emerging. The global population is immunologically naïve to this emerging pathogen, and society, at the time, had no available specific mitigation measures including immunological therapies, other than hand washing, social distancing, mask wearing and isolation when ill. Clinical and support staff in hospitals, healthcare and social care staff in care homes and other settings and key workers in

transport and infrastructure industries were at increased risk of contracting the disease[1] due to frequent contact with people with high viral loads and the high aerosolised and fomite transmission potential of this virus.[2] Early analyses in England and Wales identified main risk factors for death from COVID-19 including older age, deprivation and comorbidities[3] but did not consider how this risk differed from historical all-cause mortality among these groups. This baseline comparison enhances understanding of what additional risk is posed by COVID-19 and to whom. Excess mortality from COVID-19 in the UK has been modelled, controlling for underlying conditions and age,[4] and some conditions such as uncontrolled diabetes and severe asthma are associated with death.[3] However, understanding of the health loss impacts of COVID-19 is still limited by a lack of contextualising information, reducing our ability to respond to the challenges this disease poses, both directly and indirectly, in a proportionate, targeted manner.

We provide context for deaths and disease from COVID-19, by comparing these against a historical benchmark of when, who and how people become ill and died pre-COVID-19. Examining associations between poor COVID-19 outcomes, demographic and socioeconomic differences, age, sex and comorbidities in the context of 'usual' population health structures enhances understanding of specific risk groups and hence has a role to play in maximising the effectiveness of risk mitigation strategies while minimising the likelihood of unnecessary and undesirable impacts. Examining excess deaths (above that normally expected at a time point) is, furthermore, important for interpreting the total impact of a pandemic. Syndromic surveillance describing clinical symptoms and healthcare use is scrutinised to supplement clinical surveillance information used to populate the COVID-19 epidemic curve.

## METHODS
### Data sources
The principal data source was the Office for National Statistics (ONS); dashboards from Public Health England and syndromic surveillance in England via NHS 111 were additionally consulted. Primary focus for the analyses presented was the ONS data, which provide gold-standard confirmed recorded causes of death for UK residents. The use of ONS data is licensed under the Open Government Licence v.3.0.

### Statistical methods
All data must be viewed in their proper context before patterns can be inferred and, in this setting, against a historical baseline. In each case, profiles for COVID-19 deaths were considered against systematic differences in historical disease rates from appropriate comparison populations, to identify when disease was in excess of expected rates.

Causes of death were defined using the International Classification of Diseases, 10th Revision (ICD-10).[5] Deaths involving COVID-19 were defined as those with an underlying cause, or any mention, of ICD-10 codes U07.1 (COVID-19, virus identified) or U07.2 (COVID-19, virus not identified) on the death certificate. All causes of death are the total number of deaths registered during the same time period including those involving COVID-19.

The baseline comparison group to examine weekly temporal variation in COVID-19 deaths was deaths from respiratory disease across a historical 5-year period (2015 to 2019 inclusive). The mean number of respiratory deaths in weeks 1 to 16 of the year, together with an approximate 95% confidence interval (CI), was calculated and plotted against the numbers of COVID-19 deaths across this same time period in 2020.

When considering data regionally, rates of death per million population were the primary focus: this allowed for different population sizes within regions and hence created a metric that is comparable across geographies. Here, the mean number of deaths per million population across the previous 5-year period was used as the baseline comparison. Deaths associated with COVID-19 and excess deaths (deaths that do not attribute COVID-19 on the death certificate) were both reported.

Rates were again used to compare the risks associated with different Standard Occupational Categories (SOCs) for individuals between 20 and 64 years of age. Age-standardised rates per 100 000 population, standardised to the 2013 European Standard Population, were used in each category to correct for different numbers of people from different age groups working in each group, to ensure comparability between groups. Again, the focus is on the early part of the pandemic, with deaths registered up to and including 20th April 2020 constituting the data. Comparison with deaths from all causes occurring in these categories within the same timeframe creates a natural baseline for deciding how the rate of people dying with COVID-19 in a certain SOC compares with the rate of death in general in that SOC and helps to distinguish specific COVID-19-related effects from more subtle societal impacts, which might be influencing death rates more generally. Approximate 95% CIs were provided to facilitate comparisons.

To examine the effects in black, Asian and minority ethnic (BAME) groups, again, early pandemic data from 2nd March to 10th April 2020 inclusive were considered. Odds ratios (ORs) are used to compare categories; these were calculated by the ONS using logistic regression models, which correct for age (in 5-year categories), rural or urban inhabitants, Index of Multiple Deprivation (IMD) decile, socioeconomic status and self-reported health and activity status. Forest plots were used to show the estimated ORs for dying in each category; 95% CIs were also represented.

Finally, the representation of a panel of main comorbidities among COVID-19 deaths in March and April 2020 was explored graphically using a stacked bar

chart of the proportion of males and females separately reporting each comorbidity across age categories (including 0– to 44 years, 45–49 years, 50–54 years, 55–59 years, 60–64 years, 65–69 years, 70–74 years, 75–79 years, 80–84 years, 85–89 years and 90+ years). This allows immediate comparison of how the profile of these comorbidities changes in general by age, whether different comorbidities are more readily apparent in males and females and whether the evolution of comorbidities as age increases differs for the two sexes.

All statistical analyses were conducted in Microsoft Excel and the R statistical software package (http://www.r-project.org),[6] making use of the graphics package ggplot2.[7]

## RESULTS

### Temporal variation in COVID-19 deaths

The number of deaths from all causes varies annually and seasonally, peaking in winter. Typically, respiratory deaths range from 10% to 22% of all deaths and are seasonal, peaking annually in January; the 2015 all-death peak was high (16 237 deaths in week 2 compared with an average of 12 277 deaths that week over the previous 5 years) due to a severe influenza season, and 2018 similarly had a severe influenza season resulting in a high death count. The minimum number of weekly deaths over the previous 10-year period was 6606 (week 54, 2013). In 2020, deaths from respiratory infections were lower than the mean in the previous 5 years until early April (week 14), after which they became higher than historical rates when including deaths from COVID-19 (figure 1A) (in week 14, observed respiratory deaths exceeded the 5-year historical upper 95% CI limit by 146). An excess of unexplained deaths becomes clear

from week 14 onwards (figure 1B). Following a period of excess deaths, in week 25 of 2020, for the first time, there were fewer deaths than the equivalent previous 5-year average (65 fewer deaths), and similarly, in weeks 26 to 28, there were 917 fewer deaths than the total of the averages across years for those weeks in the previous 5 years.[8]

### Regional differences in COVID-19 deaths

Regions of England and Wales experience different death rates[8], and this pattern is true for deaths from COVID-19 (figure 2); for example, rates were highest and peaked in week 17 in London (204 per million), the North West (185 per million), the North East (179 per million) and the West Midlands (169 per million). Peak rates were lowest in the South West (95 per million) and the East Midlands (116 per million). From weeks 13 to 18 (23rd March to 3rd May), all regions of England and Wales experienced excess non-COVID-19-related deaths. This was most apparent in the West Midlands in week 17 (starting 20th April), with a peak of approximately 91 deaths per million. Between weeks 13–18 (23rd March to 3rd May), there were 46 594 excess deaths in England and Wales, 13 399 of which were listed as non-COVID-19-related. In week 25 (19th June), the total deaths dipped below the 5-year historical average for the first time (9339 compared with 9404), and this pattern continued until 10th July.

### Occupational differences in COVID-19 deaths

After age standardisation (rates per 100 000 population, standardised to the 2013 European Standard Population), men employed in low-skilled occupations (21.4, 95% CI 18.6 to 24.2) (figure 3A) were more likely to die of COVID-19-related illness (k=225 deaths from n=1321

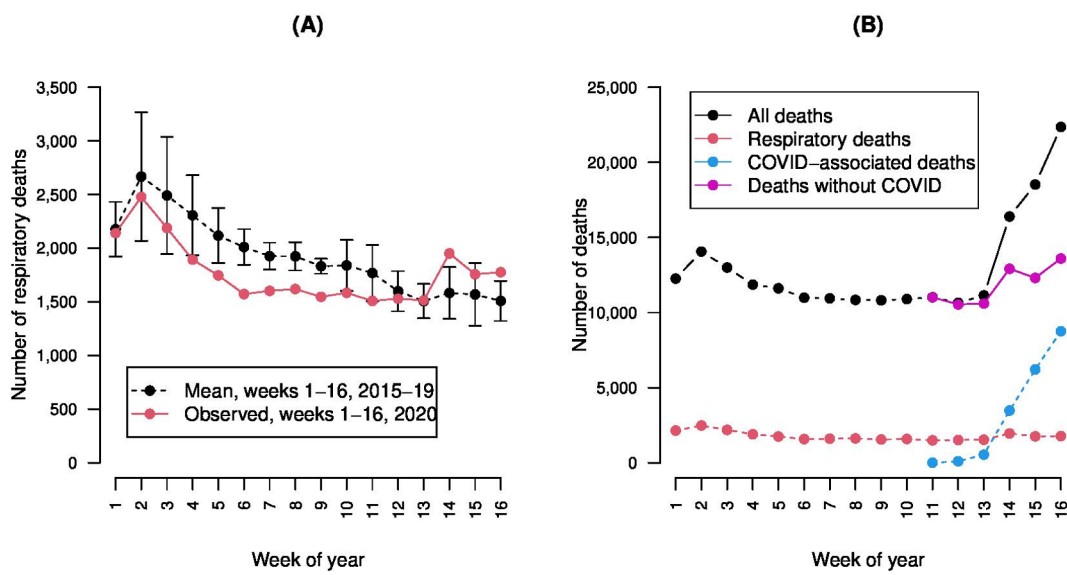

**Figure 1** Time series plots representing ((A) number of respiratory deaths per week in the first 16 weeks of 2020, by comparison with a temporally equivalent 5-year historical baseline mean (with 95% CIs), and ((B) number of deaths, respiratory deaths, deaths with COVID-19 on the death certificate and deaths without COVID-19 on the death certificate, across the first 16 weeks of 2020. Source: Office for National Statistics licensed under the Open Government Licence.

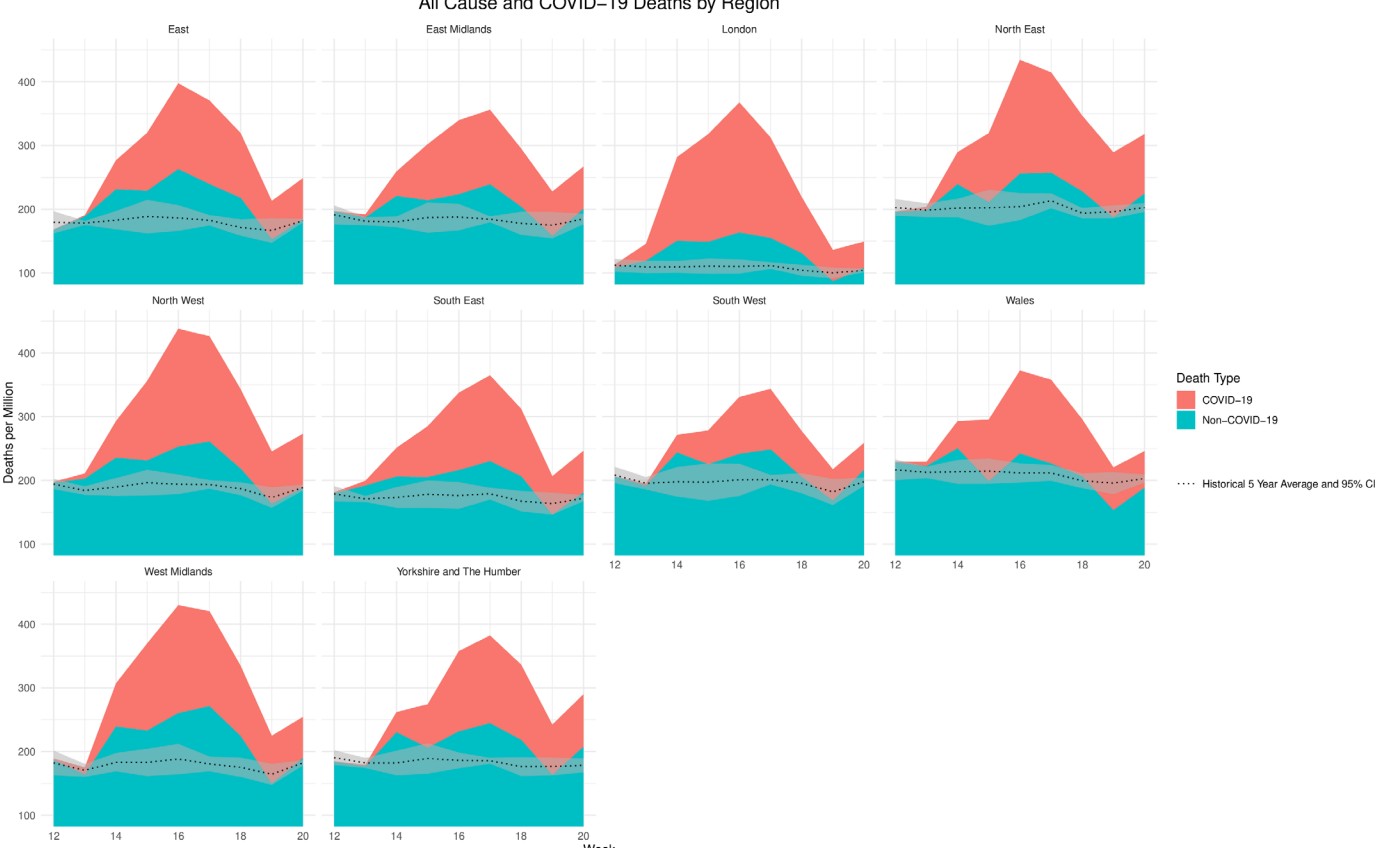

**Figure 2** All-cause and COVID-19 deaths by region between weeks 12 and 20 of 2020. Source: Office for National Statistics licensed under the Open Government Licence.

deaths in total across occupations for men[9]), as was true for all-cause mortality (figure 3B, k=915 deaths out of n=5627 deaths). This differs for women, where those employed as carers in healthcare and social care, leisure and other service operations (figure 3A) were most likely to die from COVID-19-related illness (k=130 deaths out of n=531 deaths in total across occupations for women), but not more likely to die if examining all-cause mortality (figure 3B, k=651 deaths from n=3003 deaths). For both men and women, the less technical and more manual their occupation (using ONS SOC 2020 categories), the greater the risk of dying from any cause including

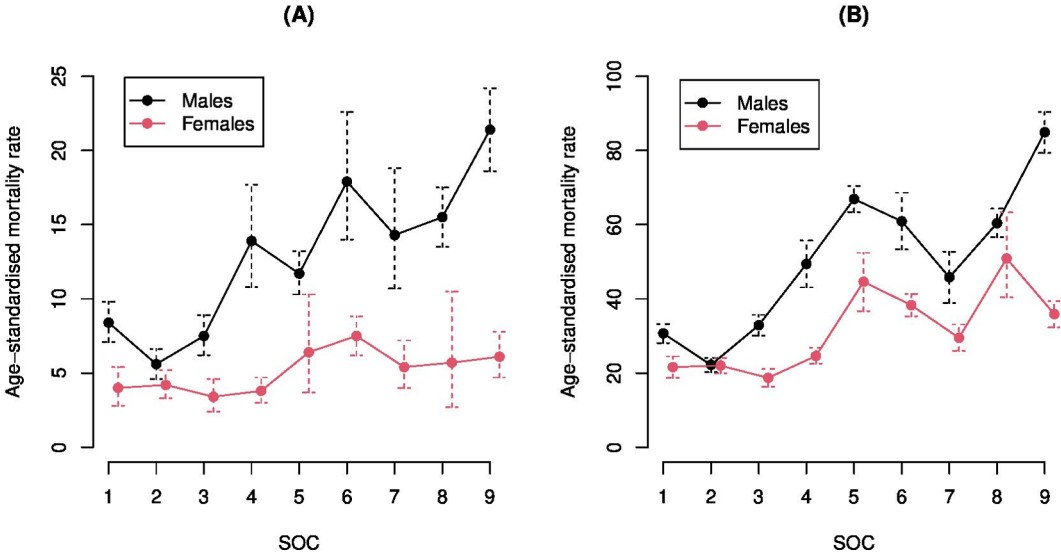

**Figure 3** Age-standardised mortality rate by Standard Occupational Category (SOC) for (A) deaths mentioning COVID-19 on the death certificate and (B) all deaths registered up to and including 20th April 2020. Source: Office for National Statistics licensed under the Open Government Licence.

**Table 1** Ratio of estimated age-standardised mortality rates comparing occupational categories with a baseline of managerial workers (SOC group 1) for (A) COVID-19-associated male and female deaths and (B) all-cause male and female deaths (including COVID-19). Source: Office for National Statistics licensed under the Open Government Licence.

| SOC | Group | Females | | Males | |
|---|---|---|---|---|---|
| | | COVID-19 | All | COVID-19 | All |
| 1 | Managers, directors and senior officials (baseline) | – | – | – | – |
| 2 | Professional occupations | 1.05 | 1.02 | 0.67 | 0.72 |
| 3 | Associate professional and technical occupations | 0.85 | 0.87 | 0.89 | 1.07 |
| 4 | Administrative and secretarial occupations | 0.95 | 1.14 | 1.65 | 1.61 |
| 5 | Skilled trade occupations | 1.60 | 2.06 | 1.39 | 2.18 |
| 6 | Caring, leisure and other service occupations | 1.88 | 1.77 | 2.13 | 1.98 |
| 7 | Sales and customer service occupations | 1.35 | 1.37 | 1.70 | 1.49 |
| 8 | Process, plant and machine operatives | 1.43 | 2.36 | 1.85 | 1.97 |
| 9 | Low-skilled elementary occupations | 1.53 | 1.66 | 2.55 | 2.77 |

Caution must be exercised in interpreting the values in table as they do not contain measures of uncertainty.
SOC, Standard Occupational Category.

COVID-19-related disease. In addition, when occupations are more manual, variation in age-standardised mortality rates is higher, particularly for men or women in certain SOC categories, for example, men undertaking administrative and secretarial roles; women in skilled trades; men in caring, leisure and other service occupations; men in sales and customer service roles; women working as process, plant and machine operatives; and men undertaking low-skilled elementary roles.

A crude comparison suggests that age-standardised mortality rates for most occupations are reduced by COVID-19 relative to deaths from any cause (table 1), with the rate only increased (for both sexes) in caring, leisure and other service occupations.

### Ethnic associations with COVID-19 deaths

As previously reported by the ONS[10] in data from 2nd March to 10th April 2020, there were increased odds of dying from COVID-19 for Bangladeshi/Pakistani (386 deaths), black (766 deaths) and Indian (483 deaths) ethnic groups (for both sexes) when compared with a baseline white group and adjusted for age, region, rural/urban, IMD decile, household composition, socioeconomic status and underlying health conditions (figure 4A,B). In total across all groups, in this time period, 12 805 deaths occurred. For Chinese and mixed ethnic groups, the OR was not statistically significantly different from one, perhaps due to small sample size (59 deaths in total observed in Chinese ethnic groups in this time period, from a total of 12 805 across all categories).

### Impact of comorbidities on COVID-19 deaths

Deaths related to COVID-19 reflect broad underlying patterns, with more reported in men (at week 15, 61.3%, n=6342) and older people (at week 15, 87% (n=8985) of deaths were in those aged over 65; 69% (n=7135) were

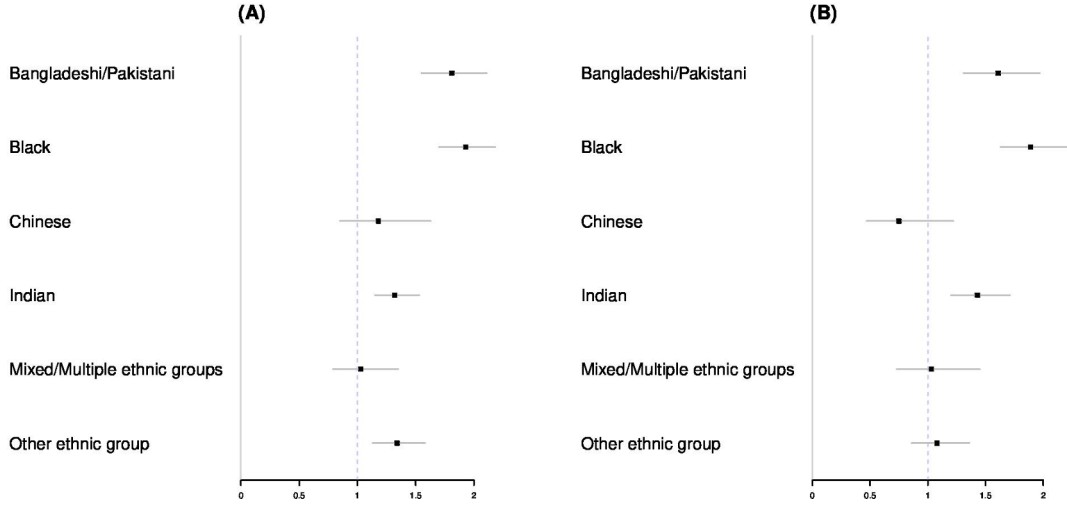

**Figure 4** ORs by ethnic category for deaths between 2nd March and 10th April 2020, which mention COVID-19 on the death certificate. Figure (4A) represents data for males; figure (4B) represents data for females. Source: Office for National Statistics licensed under the Open Government Licence.

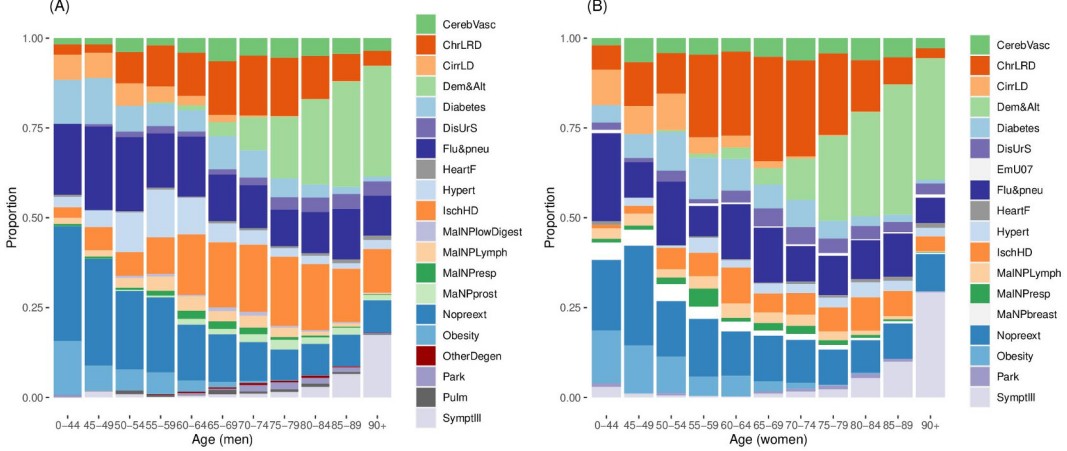

**Figure 5** From a panel of main pre-existing conditions associated with deaths from COVID-19, the proportion of patients who died with each pre-exiting condition by agegroup, based on deaths occurring between March and April 2020. CerebVasc, cerebrovascular disease; ChrLRD, chronic lower respiratory disease; CirrLD, cirrhosis and liver disease; Dem&Alt, dementia and Alzheimer's disease; diabetes; DisUrS, disease of the urinary system; Flu&pneu, influenza and pneumona; HeartF, heart failure; Hypert, hypertension; IschHD, ischaemic heart disease; MaNPbreast, malignant neoplasm of the breast; MalNPlowDigest, malignant neoplasm of the lower digestive tract; MalNPLymph, malignant neoplasm of the lymphatic system; MalNPresp, malignant neoplasm of the respiratory system; MaNPprost, malignant neoplasm of the prostate; Nopreext, no pre-existing condition; obesity; OtherDegen, other degenerative disease; Park, Parkinson's disease; Pulm, pulmonary disease; SymptIll, ill-defined symptoms. Source: Office for National Statistics licensed under the Open Government Licence.

in people aged over 75). Data from the ONS across 2019 show an increased proportion of health conditions (chest and breathing issues and heart/blood pressure/circulatory problems) related to age. The percentages of people with heart, blood pressure or circulatory problems were 0.48% (16–19 years), 6.31% (20–39 years), 31.35% (40–59 years) and 61.86% (60+ years). Similarly, the percentages of people with chest and breathing problems were 4.85% (16–19 years), 26.96% (20–39 years), 32.23% (40–59 years) and 35.96% (60+ years).[11] Long-term comorbidities such as ischaemic heart disease and hypertensive disease are commonly present in men dying with COVID-19[12] (figure 5), particularly in higher age groups; a similar pattern was observed for cerebrovascular diseases in women (figure 5B). As people reach very advanced age, for both sexes, the predominant comorbidities are dementia and Alzheimer's disease (figure 5A,B).

### Impacts of our response to COVID-19

Numerous other resources can provide information about the impacts of the human response to the pandemic. The response to COVID-19 appears to indirectly increase non-COVID-19 mortality by reducing healthcare-seeking behaviour: a big reduction in the number of visits to accident and emergency units (from 120356 in the week commencing 16th March to 89584 in the week commencing 23rd March)[13 14] coincides with the increase of both COVID-19-related and non-COVID-19-related UK deaths. There are wide impacts on a range of non-communicable diseases: for example, Cancer Research UK has estimated that for every week that routine screening is paused, 7000 people miss referrals for further tests, and 380 cancers are not diagnosed using routine screening programmes[15]; they

additionally estimate that 290000 people fewer than usual have been referred for further tests.

Data suggest that routine preventive screenings, cancer treatments, dental visits and vaccinations have all been paused to some extent during the lockdown. Evidence for this is provided in a report to the NHS by Medefer, reported in the Times (10th May 2020). At that time it suggested that by October 2020, approximately 7.2 million people would be on NHS waiting lists. The report estimates that 1.3 million people may already have been added to a lengthy waiting list, which stood at 4.4 million people in February 2020.

### DISCUSSION

This analysis characterises the early COVID-19 pandemic in England and Wales in the context of excess death over time, by region, and risk factor. Increases in mortality in April were predominantly driven by COVID-19, but non-COVID-19 excess deaths also increased in April–May 2020 across all regions. Compared with historical rates of death among occupational groups, COVID-19-related deaths generally followed normal patterns, excepting individuals among caring, leisure and other service occupations who were more likely to die from COVID-19-related illness than die from any illness. Rates of death from COVID-19-related illness are higher among BAME populations, but small sample sizes preclude all-cause mortality comparisons. Finally, pre-existing comorbidities are a strong risk factor for COVID-19-related death and are more common among men and the elderly, partly explaining why these groups appear to be at excess risk of death related to COVID-19. Thus, patterns of death and

excess death from COVID-19 mirror historical trends in mortality. This contextualisation of COVID-19 is critical to inform plans to protect the vulnerable while helping low risk populations in society to resume more normal lifestyle patterns.

The lower-than-expected death toll from week 25 onwards may be suggestive of a mortality displacement ('harvesting') impact; a proportion of the population who died at the epidemic peak (weeks 13–18) may have died in the shorter term in other circumstances. The complete picture is likely to be far more complex, but the harvesting phenomenon is previously described, for example, due to impacts of heatwaves and cold spells[16] and influenza in 1918/1919 (compared with deaths from tuberculosis).[17] Such population readjustments need to be taken into account in planning processes as the overall health loss may be relatively small compared with a disease or health problem that kills people who are healthy.

### Context to age and gender

Much age-related and gender-related health risk is more appropriately attributable to increased prevalence of underlying comorbidities. We are more likely to die as we age, with 84% of annual deaths in people over 65 years and 66% in those over 75 years.[18] Men also die earlier in most age groups and have lower life expectancies (79.2 years) than women (82.9 years).[19] As we age, our likelihood of having long-term illness increases as has been discussed. Though the burden of risk from COVID-19 lies with older age groups, more thorough epidemiological analysis may identify some subpopulations that could be classified as lower (or higher) risk. Such analysis would inform better risk management strategies, allowing mobility and economic activity among some low-risk older populations, as well as intrinsically low-risk groups such as young people.

### Context to comorbidities

Patterns of comorbidities for COVID-19-related deaths mirror the increase in these diseases with age (in non-COVID-19 circumstances); for example, ischaemic heart disease is more frequently experienced with age by men than women.[20] It is unclear whether an increasing representation of dementia and Alzheimer's as comorbidities is seen because they are genuine comorbidities in their own right or due to data biases. The most important other comorbidities are chronic lower respiratory disease in females and ischaemic heart disease in males. A role for specific genes linked to dementia and Alzheimer's and poor COVID-19 response has recently been suggested[21] and warrants further investigation. It has not been possible to know for 2020 the numbers of deaths by each comorbidity in its own right: these would be useful for comparison and establishment of any excess, but whether an excess of deaths with COVID-19 by any of the comorbidities will occur is unclear at the time of writing.

### Context to ethnicity and occupation

Ethnicity and occupation are common risk factors for morbidity and mortality from infectious disease but are not often reported in surveillance data.[22] Heightened reported risks among specific ethnic and occupational groups are alarming, and COVID-19 has brought renewed attention to health disparities inherent in the UK population, but excepting care, service and leisure workers, the precise nature and drivers of excess COVID-19 risk in different groups remain unclear. When considering occupational risk, for example, age-standardised mortality ratios (ASMRs) in different occupational categories for COVID-19 mortality must be considered alongside ASMRs for all-cause mortality. For example, when the COVID-19-associated ASMR in an occupational category is high relative to deaths from all causes, this suggests COVID-19-associated impacts should be considered in managing return to work.

### Consequences of COVID-19 and our response to its presence

The reduction in accident and emergency consultations is inconsistent with the pattern observed in 2019; it suggests a reluctance or inability of the public to access healthcare during lockdown. Unfortunately, comparisons against a longer range of historical data are not possible since the surveillance system changed in 2018, with greater numbers of hospitals reporting to the system from this point onwards. The reasons for this reduction may be multifactorial reflecting reluctance, fear of the virus and logistical difficulties for general practitionerss. This pattern of reduced healthcare uptake foreshadows an increased health burden as a result of the combination of delays introduced into the system by aspects of both the health services and individuals' responses to COVID-19. However, in the immediate future, a dip in mortality is occurring, compared with baseline. In Wales, where the median age is higher than in any other UK nation or region of England,[23] the rate of death per million returned to, or below, historical levels before any other region in England. This suggests that for high-risk populations (eg, the elderly), deaths have been compressed within the time window of the pandemic. This phenomenon was previously observed among patients with tuberculosis in the months and years following the 1918 Spanish Influenza.[17] Thus, continued contextualisation of deaths is critical to accurately assess the long-term impact of COVID-19 on health in the UK—volatility of demand should be considered in resource planning.

### Solutions: role of surveillance and need for better data reporting

What tools do we have to look at whether changes in illness patterns might be helpful in planning a response to an emerging situation such as COVID-19? ONS data are among the most accurate but have limited usefulness for real-time analysis. It is crucial that information from multiple sources is synthesised and scrutinised simultaneously, balancing timeliness against accuracy. Many

readily available sources can be used in combination to inform the evidence base. In other illnesses such as influenza,[24] a primary circulation in children may precede a secondary epidemic in the wider population. Of relevance to COVID-19 is syndromic surveillance reporting, which illustrated a spike in consultations for influenza-like illness in the under 15s above baseline for weeks 49–51 of 2019.[25] This, considered in tandem with other syndromic surveillance data, which indicated increased trips for influenza-like illness to accident and emergency units in the same period,[13] has the potential to alert society to anomalies earlier than the documented timescale for the COVID-19 pandemic. Combined scrutiny of such sources is useful to identify anomalous patterns, triggering a public health response. For example, coincident with the first reported case of COVID-19 in the UK, calls reporting cough or cold/influenza and diarrhoea spiked and then fell when the NHS 111 changed their call triage system.[25] Ensuring the comparability of age categories across reporting systems and reporting data openly at the highest resolution, which respects patient anonymity, aid rapid responsive production of understanding from research. On the international stage, authors in the USA have identified analogous issues with non-integrated reporting systems; they developed an 'App' that attempts to address some of the issues.[26] In Europe, two surveillance strands are followed, and both are restricted access: European Union/European Economic Area Member States and the UK report for every 24-hour period of the number of laboratory-confirmed cases of COVID-19 using their Early Warning and Response System. Enhanced surveillance has also been put in place via the European Surveillance System—TESSy.[27] The restricted access nature of these resources limits their real-time applicability for parties other than those with permitted access. A full consideration of the international picture is beyond the scope of this paper, but the process described herein could be repeated for other populations.

## Solutions: a model for success

Taiwan provides perhaps the best example of success in rapidly containing and controlling COVID-19; they eliminated the virus by April 2020 without going into lockdown, with minimal economic damage and few deaths.[28] Taiwan's plan for success against COVID-19 can be summarised in four points.[29] (1) In response to previous experience with SARS in 2003 and influenza H1N1 in 2009, Taiwan had developed highly functional pandemic response plans and infrastructure that were immediately operationalised in early 2019, including a Central Epidemic Command Centre and community surveillance system. (2) Taiwanese officials were quick to respond to the earliest whistle-blower reports from China with significant travel restrictions and activation of pandemic response plans. (3) The Taiwanese government is trusted and was able to successfully balance government oversight with regional autonomy. Localities and private establishments were trusted to run their own

track and trace systems, which were designed to be easily linked up to provide national coverage. Privacy concerns are acknowledged and managed, but the proven results obtained drive high levels of participation. (4) There was high buy-in from civilians across all aspects of disease control. Civilians are given space to provide suggestions and concerns in online town halls. Civilians are provided with adequate monetary support while quarantining but also face large fines, leading to high compliance. While there are cultural, social and geographical differences between the UK and Taiwan, many of these actions could be successfully deployed in the UK.

The Centers for Disease Control and Prevention (CDC) specifies a series of steps to be followed in investigating and responding to an outbreak. figure 6 outlines where this research contributes to that process and how it feeds into the wider process of outbreak management. It is clear from this figure how timely data from a variety of sources, at closely aligned degrees of temporal and spatial resolution, would streamline public health processes, significantly enhancing capacity to respond to future pandemics.

## Methodological limitations

Any analysis based on surveillance data is subject to limitations. Biases in surveillance data are well known and well documented.[30] Data on cases of disease are informative but can be heavily biased by who appears in the system and why. For example, any estimate of the case fatality ratio for COVID-19 from the early part of the pandemic would potentially be overestimated as a consequence of the likely huge underascertainment of disease in the early stages, when knowledge about COVID-19 was evolving and testing was largely limited to hospital cases of disease (the most severe manifestations). It is for this reason that the research in this paper has focused on data from the ONS, which records conclusive cause of death and is the most complete and accurate resource for UK deaths, which should ensure that any biases of reporting are minimised.

The analysis presented here is largely descriptive, and as such, it is not possible to make any statements about, for example, statistical significance of observations. This approach is deliberate: it is the authors' intention to demonstrate how a well-chosen graphical display can provide valuable insight, which can be readily interpreted by those without specialist knowledge.

## CONCLUSIONS

Policy makers have relied on models in the early phase of COVID-19. These must be supported by data-driven evidence on when, where, who and why people get sick and die. Timely emergence and analysis of this information should be used to calibrate social, cultural and economic assessments of the impact of COVID-19 versus our actions to control it, if we are to return to a cautious normality.

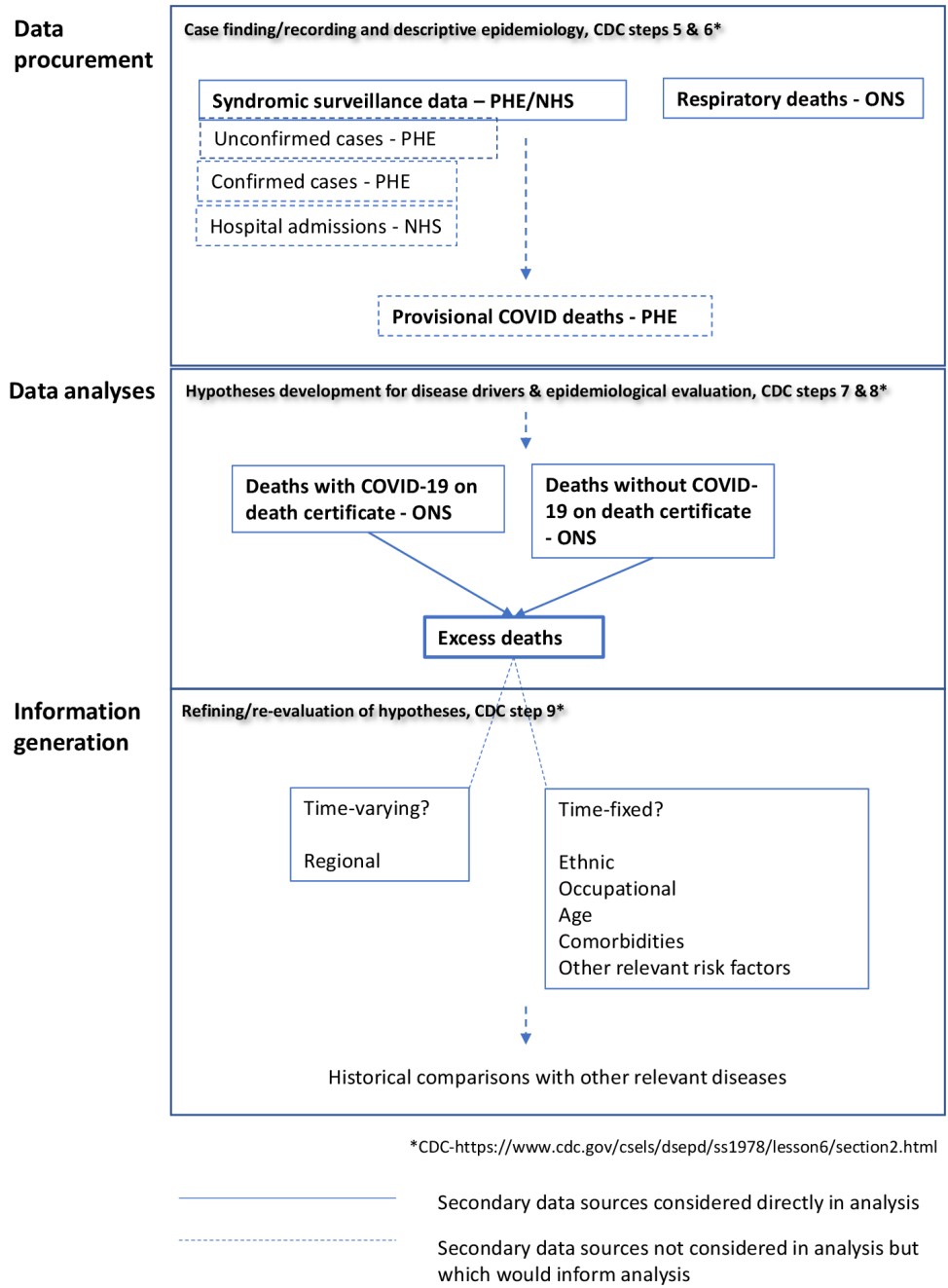

**Figure 6** Schematic describing the role of data, analysis and information generation in an iterative approach to pandemic management and infectious disease public health. CDC, Centers for Disease Control and Prevention; NHS, National Health Service; ONS, Office for National Statistics; PHE, Public Health England.

To our knowledge, this is the first study to consider reported numbers of COVID-19 illnesses and deaths in England and Wales against their historical disease context, from this variety of perspectives. Our research identifies and combines important, open-access data to inform a more nuanced response to emerging disease. Many openly available resources could improve response planning for emerging disease situations such as COVID-19 and could be used to anticipate wider consequences than immediate infection-related impacts. Syndromic surveillance data combined with real-time surveillance would supplement and strengthen the mathematical models informing emerging disease responses. Our analysis highlights the importance of calibrating social, cultural and economic assessments of the direct impact of COVID-19 against potential control actions.

**Contributors** All authors (HEC, KMM, GEP, JPH and JR) contributed equally to the planning and development of the research including its conception and design, data acquisition and analysis and interpretation of data. HEC and GEP led the writing of the manuscript. Several authors wrote and implemented the R code to produce analysis and figures: HEC (figures 1, 3, 4 and 6), GEP (figure 2) and KMM (figures 5 and 6).

**Funding** The work was funded through the employment of the authors at the University of Liverpool, UK.

**Competing interests** None declared.

**Patient and public involvement** Patients and/or the public were not involved in the design or conduct or reporting or dissemination plans of this research.

**Patient consent for publication** Not required.

**Provenance and peer review** Not commissioned; externally peer reviewed.

**Data availability statement** Data are available in a public, open-access repository. Data and reports consulted in this study are in the public domain. The principal data source was the Office for National Statistics (ONS), which provides gold-standard confirmed causes of death for UK residents; dashboards from Public Health England (PHE) and syndromic surveillance in England via NHS 111 were additionally consulted. Use of ONS data is licensed under the Open Government Licence v.3.0, and data are available from https://www.ons.gov.uk/; Public Health England dashboard is located at https://coronavirus.data.gov.uk/, and NHS 111 reports are available via https://digital.nhs.uk.

**ORCID iD**
Helen E Clough http://orcid.org/0000-0002-8538-0684

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
