## [Reviewer comments · BMJ Open]

ARTICLE DETAILS

TITLE (PROVISIONAL)	Use of routine death and illness surveillance data to provide insight for UK pandemic planning: lessons from COVID-19
AUTHORS	Clough, Helen; McIntyre, Kirsty; Patterson, Grace; Harris, John; Rushton, Jonathan

VERSION 1 – REVIEW

REVIEWER	Prof Kiran Shekar The Prince Charles Hospital Brisbane Australia
REVIEW RETURNED	27-Sep-2020

GENERAL COMMENTS	Thank you for the opportunity to review this important study To start with let me clarify, I am reviewing this paper as a clinician scientist and I do not have specific expertise in public health or social medicine. The data is from open sources. We need to focus on trends here as the total burden of cases and attributable deaths may be over or underestimated. Also, these patterns may or may not be replicated in other jurisdictions. I have no reason to suspect the integrity of data as such, but given the potential impact that this paper may have on the media, public and political discourse, I request the editors to independently verify the data and statistical analysis. This is a sobering analysis and I congratulate the authors on their effort. I make these observations mainly to articulate my thoughts after reading this nice paper and make couple of suggestions in between to consider. I have no specific comments other than being intrigued what the data for the whole year 2020 would look like and I strongly recommend that the authors follow up on this for the greater good as there is so much to be learnt. Our approach has to be more sophisticated than generating computer models that say how many are likely to be infected or die in a country while placing the whole population at similar risk of death and having non-targeted mitigation measures. A more nuanced analysis such as this can give more purpose and meaning to our efforts of protecting the vulnerable better. This is pretty much the story in every country and the partisan nature of politics, leadership failures and policy approaches have left us all divided and people are conveniently and selectively using data to support their arguments. Hope analysis such as this may help us reshape our policy and public health approaches to second and subsequent waves of the COVID-19 pandemic and also for future pandemics.
--

	This paper will carry greater meaning if authors articulate what an ideal public health policy approach may look like based on their data and lessons learnt - even a Table or a figure/animation to get the message across may be quite helpful to the reader- this may allow a more constructive debate around this paper The socioeconomic inequities lead to variable baseline health and access to health care in our general population at best of times. This is then compounded by fear to access health care in a pandemic leading to excess COVID and non-COVID deaths in some populations. This is a sad reality of the world we live in and of the systems we built. How do we compensate for these inequities .. some people are simply born to live with progressive chronic disease, accumulate greater risk for an acute illness over a lifetime and die a premature death regardless of a pandemic or not -based on this paper and other published body of literature we clearly know who those disadvantaged people are, question is what do we do about it ? Individual responsibilities and accountabilities matter when it comes to one's personal health and wellbeing and no denying that, but there are other factors that make some populations accumulate greater burden of comorbidities. If large sections of society are at risk (if not for COVID for something else) mainly not because of the genes they are born with, but because of the modifiable and non-modifiable inequities/circumstances that are so deeply embedded in our societies, even the best pandemic response may not fix this .. this needs an inter-generational fix Hope when the next once in a century pandemic strikes, we have a more equitable and better prepared world that will crush it and hope a similar data analysis then will be less confronting.
--	--

REVIEWER	Mitra Amini Shiraz University of Medical Sciences, Shiraz, Iran
REVIEW RETURNED	15-Oct-2020

GENERAL COMMENTS	Dear author Thank you for submitting your valuable manuscript to BMJ open. There are some minor issues: You used the abbreviation in the abstract such as ONS, NHS, BAME, PHE. It is better to mention the words completely and then put the abbreviation in the parentheses. The readers of the BMJ open are international readers, I think you can summarize some parts and instead add a comparison between your data and other data from all over the world.
--

REVIEWER	Jonathon P Leider University of Minnesota, Minnesota, US
REVIEW RETURNED	19-Oct-2020

GENERAL COMMENTS	The manuscript covers an extremely important topic in excess mortality in the UK. The major issues I take are that, from the perspective of a research paper, the authors do not do an adequate job describing methods, parsing results from discussion, or identifying potential limitations. The methods section needs to be completely reworked and expanded. Even if you are doing merely descriptive reporting (which is appropriate given the data), you don't talk about how you do big things (e.g., age-standardization) or small things (e.g., adjusting for the fact that 2020 is a leap year). Are you age standardizing by race/ethnicity
---

	and gender? By geography? It is not clear. It would be my expectation that this would be a much more substantive section. The results and discussion section are not appropriately delineated right now. The results should have the actual, novel results and discussion should handle implications of those results. Right now, they are quite mixed. There are a number of items you discuss that are not talked about in the results. Agree you should look more into BAME outcomes versus white outcomes. Incredibly important.
--	--

REVIEWER	Ko, Yi-An Emory University
REVIEW RETURNED	09-Nov-2020

GENERAL COMMENTS	The study methods and statistical methods are lacking. No details were provided for others to replicate the analysis and the results given the same data. What were the inclusion criteria (e.g., dates, regions, etc.) to obtain the analysis data sets? How were COVID diagnosis and COVID-related deaths confirmed? The abstract mentions death certificate but nowhere in the text described this with further details. How was occupation defined and the what was the source (e.g., people can switch jobs at any time)? What statistical software was used? What was the statistical significance level? Figure legends should be provided with explanations/definitions for all the abbreviations used in the graphs. Table 1 shows age-standardised mortality rates. How was this calculated? If regression was used, what was it and how was age treated in the analysis (categorical, linear continuous variable, or others)? Figure 1 shows the number of deaths comparing 2020 vs previous years (2010-2019). 2020 appears to have the lowest number of deaths in the last 10 years. So, there could be a temporal trend of number of deaths over the years. It would be helpful to show adjusted/expected number of deaths rather than the crude numbers if the purpose is to compare 2020 with previous years. Fig 1a and 1b can be combined. Also, Figures 1 and 2 will be more meaningful if the numbers of deaths adjusted for population size were presented. If Figure 3 is to compare mortality in certain occupations vs. overall mortality, it may be more informative to have men's data in one figure and women's in the other, or just to have one figure with four lines. What is the unit of the mortality rate? What do the intervals mean? On page 8 first paragraph, "For Chinese and mixed ethnic groups the odds ratio was not statistically significantly different from one, perhaps due to sample size." Were these tested? What is the method/model? The sample sizes should be provided for all the subgroups presented in the paper.
--

VERSION 1 – AUTHOR RESPONSE

Reviewer: 1

Comments to the Author

Thank you for the opportunity to review this important study To start with let me clarify, I am reviewing this paper as a clinician scientist and I do not have specific expertise in public health or social medicine.

The data is from open sources. We need to focus on trends here as the total burden of cases and attributable deaths may be over or underestimated. Also, these patterns may or may not be replicated in other jurisdictions.

I have no reason to suspect the integrity of data as such, but given the potential impact that this paper may have on the media, public and political discourse, I request the editors to independently verify the data and statistical analysis.

This is a sobering analysis and I congratulate the authors on their effort. I make these observations mainly to articulate my thoughts after reading this nice paper and make couple of suggestions in between to consider.

I have no specific comments other than being intrigued what the data for the whole year 2020 would look like and I strongly recommend that the authors follow up on this for the greater good as there is so much to be learnt.

Our approach has to be more sophisticated than generating computer models that say how many are likely to be infected or die in a country while placing the whole population at similar risk of death and having non-targeted mitigation measures. A more nuanced analysis such as this can give more purpose and meaning to our efforts of protecting the vulnerable better.

This is pretty much the story in every country and the partisan nature of politics, leadership failures and policy approaches have left us all divided and people are conveniently and selectively using data to support their arguments. Hope analysis such as this may help us reshape our policy and public health approaches to second and subsequent waves of the COVID-19 pandemic and also for future pandemics.

This paper will carry greater meaning if authors articulate what an ideal public health policy approach may look like based on their data and lessons learnt - even a Table or a figure/animation to get the message across may be quite helpful to the reader- this may allow a more constructive debate around this paper The socioeconomic inequities lead to variable baseline health and access to health care in our general population at best of times. This is then compounded by fear to access health care in a pandemic leading to excess COVID and non-COVID deaths in some populations. This is a sad reality of the world we live in and of the systems we built.

We have introduced a new figure (Figure 6) which outlines a generic process which should provide timely evidence to target where work needs to be undertaken in the event of an emerging situation (and associated reference within the text (lines 517 to 544)). This schematic (a) sets down what we looked at, and (b) how this sits within a wider, more complete process and (c) highlights where there are deficiencies which allow us to make suggestions for improvements. The evidence-gathering process is visualised as a system which combines data procurement, data analysis, and information generation. Clear emphasis of what has been studied in this article and what other information might be useful. An ideal public health policy would capture relevant data in real time, do timely and pertinent analysis which draws upon the range of resources outlined in Figure 6 to inform decision making. The cycle of evidence is complete by effective communication of results. Paper addresses steps 5 and 6 in the CDC stages and touches on stages 7, 8 and 9.

How do we compensate for these inequities .. some people are simply born to live with progressive chronic disease, accumulate greater risk for an acute illness over a lifetime and die a premature death regardless of a pandemic or not -based on this paper and other published body of literature we clearly know who those disadvantaged people are, question is what do we do about it ?

Individual responsibilities and accountabilities matter when it comes to one's personal health and wellbeing and no denying that, but there are other factors that make some populations accumulate greater burden of comorbidities. If large sections of society are at risk (if not for COVID for something else) mainly not because of the genes they are born with, but because of the modifiable and non-modifiable inequities/circumstances that are so deeply embedded in our societies, even the best pandemic response may not fix this .. this needs an inter-generational fix Hope when the next once in a century pandemic strikes, we have a more equitable and better prepared world that will crush it and hope a similar data analysis then will be less confronting.

Reviewer 2

Thank you for submitting your valuable manuscript to BMJ open. There are some minor issues: You used the abbreviation in the abstract such as ONS, NHS, BAME, PHE. It is better to mention the words completely and then put the abbreviation in the parentheses.

We have fully defined all of these terms at first usage (lines 47 to 49) and abbreviated thereafter.

The readers of the BMJ open are international readers, I think you can summarize some parts and instead add a comparison between your data and other data from all over the world.

A full comparison of countries is beyond the scope of this paper, but our approach is one that in principle could be repeated for other populations. Given also space constraints we have added a paragraph to the discussion describing surveillance initiatives in the US and Europe (lines 509 to 516).

Reviewer: 3

Comments to the Author

The manuscript covers an extremely important topic in excess mortality in the UK. The major issues I take are that, from the perspective of a research paper, the authors do not do an adequate job describing methods, parsing results from discussion, or identifying potential limitations. The methods section needs to be completely reworked and expanded. Even if you are doing merely descriptive reporting (which is appropriate given the data), you don't talk about how you do big things (e.g., age-standardization) or small things (e.g., adjusting for the fact that 2020 is a leap year). Are you age standardizing by race/ethnicity and gender? By geography? It is not clear. It would be my expectation that this would be a much more substantive section.

We thank reviewer 3 for their comments and in response to this have reworked this section to include dates, reference denominator populations for e.g. age standardisation (lines 173 to 231). We do not believe that 2020 being a leap year will have a huge impact for this analysis and have hence not sought to correct for this.

The results and discussion section are not appropriately delineated right now. The results should have the actual, novel results and discussion should handle implications of those results. Right now, they are quite mixed. There are a number of items you discuss that are not talked about in the results. Agree you should look more into BAME outcomes versus white outcomes. Incredibly important.

Sections have been moved and relocated accordingly (lines 345 to 353).

Reviewer: 4

Comments to the Author

The study methods and statistical methods are lacking. No details were provided for others to replicate the analysis and the results given the same data. What were the inclusion criteria (e.g., dates, regions, etc.) to obtain the analysis data sets? How were COVID diagnosis and COVID-related deaths confirmed? The abstract mentions death certificate but nowhere in the text described this with further details. How was occupation defined and the what was the source (e.g., people can switch jobs at any time)? What statistical software was used? What was the statistical significance level?

We thank reviewer for their comments and have reworked the methods section in line with the these requests (lines 173 to 231). Clearer information on the categorisation of cause of death, time-windows for data inclusion, methods of age- and other-standardisation, reference populations and software used have now been provided in this section.

Figure legends should be provided with explanations/definitions for all the abbreviations used in the graphs.

These are listed inside the resubmission.

Table 1 shows age-standardised mortality rates. How was this calculated? If regression was used, what was it and how was age treated in the analysis (categorical, linear continuous variable, or others)?

Approaches are now clarified via a greatly expanded methods section (lines 173 to 231).

Figure 1 shows the number of deaths comparing 2020 vs previous years (2010-2019). 2020 appears to have the lowest number of deaths in the last 10 years. So, there could be a temporal trend of number of deaths over the years. It would be helpful to show adjusted/expected number of deaths rather than the crude numbers if the purpose is to compare 2020 with previous years. Fig 1a and 1b can be combined.

Also, Figures 1 and 2 will be more meaningful if the numbers of deaths adjusted for population size were presented.

We appreciate the suggestion to combine Figures 1a and 1b, but we do not think that combining these will provide for effective visual display, since having "all deaths" (which operates on a much larger scale) on the graph dampens down the effects in Figure 1(a) that we intend the reader to observe. It is our view that the key messages from both graphs are more clearly represented as separate panels.

We examined the population data and found that the population size increased steadily, but not greatly in magnitude, during the period 2015 to 2019 (it has increased on average by approximately 400,000 per year; 0.6% based upon an average population size across that period). A plot of the mean number of respiratory deaths per 100,000 population together with the numbers of deaths per 100,000 reveals the same intrinsic pattern as Figure 1(a). Given that the population size has not changed hugely in this same time window, we hence believe that the inference to be drawn from our original presentation of the data does not change. Further, our information on where the deaths took place is scant: it is clear, for example, that deaths in care homes featured large, but specific care home data is not available for this time period and hence we do not know the population at risk. The population at risk varies across the country and is constantly changing: this is in the nature of an

epidemic. For these reasons we do not think we could accurately provide population corrections or greater insight. We have made one change to Figure 1(a), altering our historical comparison period to the previous five years, to bring it into line with the regional analysis presented in Figure 2. Finally, the reviewer is correct that the death toll in the earlier part of 2020 is low by comparison with the historical average and we have noted this in the comments on Figure 1a in the manuscript (line 244; and “temporal variation” section).

Figure 2 is already adjusted for population size (deaths per million).

If Figure 3 is to compare mortality in certain occupations vs. overall mortality, it may be more informative to have men’s data in one figure and women’s in the other, or just to have one figure with four lines. What is the unit of the mortality rate? What do the intervals mean?

Mortality rate definitions have been clarified in the text (lines 273 to 274).

We have tried the suggested alternative mode of display and concluded that it is not more illuminative. It becomes difficult to see where COVID risks deviate from ordinary risks as both operate on different scales and displaying them together masks any contrasting patterns, which was the point of the display we had created ((a) = males; (b) = females):

On page 8 first paragraph, “For Chinese and mixed ethnic groups the odds ratio was not statistically significantly different from one, perhaps due to sample size.” Were these tested? What is the method/model?

We have clarified sample size and methods in the appropriate sections (lines 312 to 321).

The sample sizes should be provided for all the subgroups presented in the paper.

These have been added throughout the manuscript.

VERSION 2 – REVIEW

REVIEWER	Kiran Shekar The Prince Charles Hospital, Brisbane, Queensland, Australia
REVIEW RETURNED	23-Dec-2020

GENERAL COMMENTS	thank you for the revisions. the paper is reading very well
---

REVIEWER	Mitra Amini Shiraz University of Medical Sciences, Shiraz, Iran
REVIEW RETURNED	27-Dec-2020

GENERAL COMMENTS	Dear Authors Thank you so much for your revision.
--